# Associations of A20, CYLD, Cezanne and JAK2 Genes and Immunophenotype with Psoriasis Susceptibility

**DOI:** 10.3390/medicina59101766

**Published:** 2023-10-03

**Authors:** Nguyen Hoang Giang, Nguyen Thi Kim Lien, Do Thi Trang, Pham Thi Huong, Nguyen Huy Hoang, Nguyen Thi Xuan

**Affiliations:** 1Institute of Genome Research, Vietnam Academy of Science and Technology, Hoang Quoc Viet, Ha Noi 100000, Vietnam; 2Department of Biotechnology, University of Science and Technology of Hanoi, Vietnam Academy of Science and Technology, Hoang Quoc Viet, Ha Noi 100000, Vietnam

**Keywords:** A20, Cezanne, CYLD, immunophenotype, JAK2, psoriasis

## Abstract

*Background and Objectives*: Psoriasis is an immune-mediated chronic inflammatory skin disorder and commonly associated with highly noticeable erythematous, thickened and scaly plaques. Deubiquitinase genes, such as tumor necrosis factor-alpha protein 3 (TNFAIP3, A20), the cylindromatosis (CYLD) and Cezanne, function as negative regulators of inflammatory response through the Janus kinase/signal transducers and activators of transcription (JAK-STAT) pathways. In this study, polymorphisms and expressions of A20, CYLD and Cezanne genes as well as immunophenotype in psoriatic patients were determined. *Materials and Methods*: In total, 82 patients with psoriasis and 147 healthy individuals with well-characterized clinical profiles were enrolled. Gene polymorphisms were determined by direct DNA sequencing, gene expression profile by quantitative real time-polymerase chain reaction (PCR), immunophenotype by flow cytometry, and the secretion of cytokines and cancer antigen (CA) 125 by enzyme-linked Immunosorbent assay (ELISA). *Results*: The inactivation of A20, CYLD and Cezanne and increased levels of TNF-α, IFN-γ and CA 125 was observed in psoriatic patients. Importantly, patients with low A20 expression had significant elevations of triglyceride and total cholesterol concentrations and higher numbers of CD13^+^CD117^−^ and CD19^+^CD23^+^ (activated B) cells than those with high A20 expression. Genetic analysis indicated that all rs4495487 SNPs in the JAK2 gene, rs200878487 SNPs in the A20 gene and four SNPs (c.1584-375, c.1584-374, rs1230581026 and p.W433R) in the Cezanne gene were associated with significant risks, while the rs10974947 variant in the JAK2 gene was at reduced risk of psoriasis. Moreover, in the Cezanne gene, p.W433R was predicted to be probably damaging by the Polyphen-2 prediction tool and an AA/CC haplotype was associated with a high risk of psoriasis. In addition, patients with higher CA 125 levels than the clinical cutoff 35 U/mL showed increased levels of IFN-γ than those with normal CA 125 levels. *Conclusions*: A20 expression was associated with lipid metabolism and the recruitment of CD13^+^ CD117^−^ and activated B cells into circulation in psoriatic patients. Besides this, the deleterious effect of the p.W433R variant in the Cezanne gene may contribute to the risk of psoriasis.

## 1. Introduction

Psoriasis is a chronic inflammatory skin disorder, influenced by the combination of genetic, epigenetic and environmental factors [1]. The most common type is plaque psoriasis that presents with erythematous and scaly plaques on the scalp, trunk, and limbs. The degree of systemic inflammation in psoriasis is determined by immune cell infiltration and activation, leading to a consequence of the hyperproliferation of keratinocytes and epidermal alterations [2]. Immunophenotypic studies indicated that the counts of CD4^+^T, CD19^+^B and T helper 17 cells are elevated, while the proportion of natural killer (NK) cells is decreased in the peripheral blood of psoriatic patients [3,4,5]. The activation of B and T cells in these patients leads to the release of pro-inflammatory cytokines such as interleukin (IL)-17, IL-23, interferon (IFN)-γ and tumor necrosis factor (TNF)-α [6], which causes prolonged inflammation and damage to multiple tissues and organs [4,7]. In contrast, the percentage of regulatory B cells, which are negative regulators of the immune response, is impaired in patients with psoriasis [8]. The activation of the janus kinase/signal transduction and activator of transcription (JAK-STAT) has been reported to relate to inflammatory response in psoriasis [9]. In addition, the abnormal expression of both aminopeptidase N (CD13) and Fc epsilon RIIb (CD23) has been implicated in psoriasis. CD13 is highly expressed on psoriatic fibroblasts [10], while CD23 expression is enhanced in patients with psoriasis [11]. CD23 is considered as a potential therapeutic target for inflammatory disorders [12].

Immunogenetic investigations revealed that most psoriasis susceptibility loci are linked to inflammatory response-related genes. Among them, deubiquitinase (DUB) genes, including tumor necrosis factor α-induced protein 3 (TNFAIP3, A20), tumor suppressor cylindromatosis (CYLD) and Cezanne, play important roles in deubiquitinating target proteins by cleaving their polyubiquitin chains to suppress the activation of downstream signalling pathways. The attenuated expression of A20 and CYLD is shown in patients with leukemia/lymphoma [13,14,15] and Cezanne is down-regulated in hepatocellular carcinoma [16]. A20 variants are associated with susceptibility to psoriasis in a Japanese population [17]. A20 mRNA expression level in peripheral blood mononuclear cells (PBMCs) is negatively correlated with disease severity in psoriasis [18] and A20 overexpression in keratinocytes significantly suppresses the release of inflammatory cytokines and chemokines [19]. Recently, several CYLD variants have been related to the development of autoimmune diseases and psoriasis [20]. Unlike A20 and CYLD, the effects of Cezanne on the pathogenesis of psoriasis are not well known.

Similar to A20, the association of JAK2 with psoriasis susceptibility has also been extensively studied, as JAK2 is involved in the control of cellular growth and proliferation during immune responses in these patients [21]. A20 is known as a regulator of inflammatory response through the JAK2 signaling pathway [22]. Polymorphisms in the JAK2 gene, including rs2274471 and rs7849191, are detected in Korean patients with psoriasis [21].

In this study, the expressions of A20, CYLD and Cezanne were examined by quantitative real time-PCR and immunophenotype by flow cytometry in 82 patients with psoriasis and 147 healthy individuals. Besides this, polymorphisms of A20, Cezanne, CYLD and JAK2 genes were also determined by direct DNA sequencing.

## 2. Materials and Methods

### 2.1. Patients and Control Subjects

A total of 82 untreated psoriatic patients and 147 healthy volunteers used as controls were recruited into the study at the Thien Phu Duong Traditional Medical Clinic, Hanoi, Vietnam. The diagnosis of psoriasis was based on the 2016 WHO criteria [23], including sharply demarcated round–oval erythematous plaques with loosely adherent silvery white scales, especially affecting the elbows, knees, lumbosacral area, intergluteal cleft, and scalp. All patients were aged ≥22 years with plaque-type psoriasis defined at enrolment by Psoriasis Area Severity Index (PASI) score ≥ 3. No individuals in the control population took any medication or suffered from any known acute or chronic disease. All patients and volunteers gave written consent to participate in the study. Person care and experimental procedures were performed according to the Vietnamese law for the welfare of humans and approved by the Ethical Committee of Institute of Genome Research, Vietnam Academy of Science and Technology under number 03-2021/NCHG-HĐĐĐ on 14 January 2021.

### 2.2. RNA Extraction and Real-Time RT-PCR

Total mRNA was isolated using the Qiashredder and RNeasy Mini Kit from Qiagen according to the manufacturer’s instructions. For cDNA first strand synthesis, 1 µg of total RNA in 12.5 µL DEPC-H2O was mixed with 1 µL of oligo-dT primer (500 µg/mL, Invitrogen, Waltham, MA, USA) and heated for 2 min at 70 °C. To determine the transcript levels of A20, Cezanne, CYLD and GAPDH, quantitative real-time PCR with the LightCycler System (Roche Diagnostics, Basel, Switzerland) was applied. The following primers were used: A20 primers—5′-TCCTCAGGCTTTGTATTTGA-3′ (forward) and 5′-TGTGTATCGGTGCATGGTTTT-3′ (reverse); Cezanne primers—5′-ACAATGTCCGATTGGCCAGT-3′ (forward) and 5′-ACAGTGGGATCCACTTCACATTC-3′ (reverse); CYLD primers—5′-TGCCTTCCAACTCTCGTCTTG-3′ (forward) and 5′-AATCCGCTCTTCCCAGTAGG-3′ (reverse), and GAPDH primers—5′-GGAGCGAGATCCCTCCAAA-3′ (forward) and 5′-GGCTGTTGTCATACTTCTCAT-3′ (reverse). PCR reactions were performed in a final volume of 20 µL containing 2 µL cDNA, 2.4 µL MgCl2 (3 µM), 1 µL primer mix (0.5 µM of both primers), 2 µL cDNA Master SybrGreen I mix (Roche Molecular Biochemicals), and 12.6 µL DEPC-treated water. The target DNA was amplified during 40 cycles of 95 °C for 10 s, 62 °C for 10 s, and 72 °C for 16 s, each with a temperature transition rate of 20 °C/s, a secondary target temperature of 50 °C, and a step size of 0.5 °C. Melting curve analysis was performed at 95 °C for 0 s, 60 °C for 10 s and 95 °C for 0 s to determine the melting temperature of primer dimers and the specific PCR products. The ratio between the respective gene and corresponding GAPDH was calculated per sample according to the ∆∆ cycle threshold method [24].

### 2.3. DNA Sequencing 

Genomic DNA was isolated from peripheral blood samples using a DNeasy blood and tissue kit (Qiagen, Hilden, Germany). To determine polymorphisms of the JAK2, A20, CYLD and Cezanne genes, polymerase chain reaction (PCR) and DNA sequencing (3500 Genetic Analyzers, Thermo Scientific, Waltham, MA, USA) were performed as previously described [25]. The GenBank accession numbers NM_004972.4, NM_00137874.1, NM_001270508.2 and NM_020205.4 were used for DNA sequence analysis of JAK2, A20, CYLD and Cezanne, respectively, by using primers JAK2-F: 5′- TCAAAACCGAGTAGAGCCAA-3′ and JAK2-R: 5′-GTAGAGGAGCCTCTGTGTAACC-3′; CYLD -F: 5′-TAAGGTCTTGTGCCTGAGCA-3′ and CYLD-R: 5′-TTCTTTGGCAGCAGAAATCC-3′; A20-F: 5′-TGAGCTAATGATGTAAAATCTTGTG-3′ and A20-R: 5′- AGGAGGCCTCTGCTGTAGTC-3′ and Cezanne-F: 5′-GCCTCCTGCATCAACTTCCT-3′ and Cezanne-R: 5′-TCAGAGGACAGTGGGATCCA-3. The amplification product lengths of JAK2, CYLD, A20 and Cezanne were 686, 731, 546 and 600 bp, respectively. All obtained PCR fragments were purified with a GeneJET PCR purification kit (Thermo Scientific, Waltham, MA, USA). The PCR products were sequenced on both strands by Sanger sequencing using the ABI 3500 sequencer (Thermo Fisher Scientific, Waltham, MA, USA).

### 2.4. Immunostaining and Flow Cytometry

Immunophenotypic features of PBMCs from psoriatic patients and healthy volunteers were determined by flow cytometry (FACSAria Fusion, BD Biosciences, Moscow, Russia) as previously described [26]. Cells (10^5^) were incubated in 100 µL FACS buffer (PBS plus 0.1% FCS) containing fluorochrome-coupled antibodies to CD45, FoxP3, CD19, CD13, CD23, CD117, CD3, CD4, and CD25 (all from Thermo Fisher Scientific, Waltham, MA, USA) at a concentration of 10 µg/mL. After incubating with the antibodies for 60 min at 4 °C, the cells were washed twice and resuspended in FACS buffer for flow cytometry analysis.

### 2.5. Cytokine Quantification

Sera were isolated from the blood samples of psoriatic patients and healthy subjects and stored at −20 °C until used for ELISA. TNF-α, IFN-γ and CA125 concentrations were determined using ELISA kits (Thermo Fisher Scientific, Waltham, MA, USA) according to the manufacturer’s protocol.

### 2.6. Data Analysis

Data of genes related to this record were collected from NCBI (https://www.ncbi.nlm.nih.gov/) accessed on 10 January 2022. The information for SNP ID of these genes was retrieved from the NCBI’s SNP database (https://www.ncbi.nlm.nih.gov/snp/) accessed on 6 January 2023. Bioedit software was used for the initial analysis of the sequences. To analyze the functional consequences of deleterious SNPs, a PolyPhen2 program (http://genetics.bwh.harvard.edu/pph2/index.shtml) accessed on 20 May 2023 was used. The PolyPhen-2 score varied from 0.0 (tolerated) to 1.0 (deleterious), in which the SNPs were designated “probably damaging”, “potentially damaging”, “benign” or “unknown”.

### 2.7. Statistics

Statistical analysis was performed with the SPSS version 20 (IBM, New York, NY, USA) and GraphPad Prism version 8.4 softwares (Boston, MA, USA). To analyze the data, Chi-squared (χ^2^) tests, crosstabs and odds ratios were used for dichotomous data, and the Mann–Whitney U test for nominal data. *p* < 0.05 was considered statistically significant.

## 3. Results

### 3.1. Correlation among A20 Expression, Immunophenotype and Clinical Outcomes in Psoriatic Patients

As shown in Table 1, the mean age of psoriatic patients was 36.9 years. The gender distribution in the patient group was 56 males and 26 females. The clinical profiles showed significant increases in triglyceride and total cholesterol concentrations in the patient group (Table 1). Cytokine production in the sera of psoriatic patients was also examined. Similar to a recent study [6], we observed that serum levels of IFN-γ and TNF-α in psoriatic patients were significantly higher than in control individuals (Figure 1A). Next, CA125 is known to be involved in the development of several autoimmune diseases, such as systemic lupus erythematosus and rheumatoid arthritis [27,28]; therefore, we performed experiments to examine whether its levels are increased in psoriasis patients. As expected, CA125 serum levels were significantly higher in psoriatic patients than in the control group (Figure 1B), and the number of psoriatic patients with CA125 levels above the clinical cutoff, 35 U/mL, was 36 samples (43.9%). Importantly, increasing CA125 levels were significantly correlated with higher levels of IFN-γ (Figure 1C). The evidence suggests that there was a strong correlation between the release of CA125 and IFN-γ in psoriatic patients.

Since A20, CYLD and Cezanne negatively regulate the inflammatory response in autoimmune and cancer diseases, their mRNA levels were examined by quantitative real-time PCR. The results indicate that the inactivated expression of A20, CYLD and Cezanne was found in the patient group as compared healthy controls (Figure 1D). Next, we asked whether there are associations between A20, CYLD and Cezanne expression levels and clinical outcomes. Firstly, the expression of A20 in psoriatic patients was divided into two groups based on the median A20 expression value in healthy controls (high vs. low). High A20 expression was detected in 20 samples (24.4%) and low A20 expression was detected in 62 samples (75.6%). The relationship between A20 expression and clinical features at diagnosis was analyzed. The results show that patients with low A20 expression had a significant elevation of triglyceride and total cholesterol concentrations compared to those with high A20 expression (Table 1). In addition, significant differences in other clinical indicators between the two A20 expression level groups did not exist (Table 1). Similar to A20, Cezanne and CYLD expressions were also divided into two groups based on the median Cezanne/CYLD expression values in healthy controls (high vs. low); however, we did not observe significant differences between Cezanne/CYLD expression levels and all the clinical indicators in psoriatic patients (data not shown).

Finally, immunophenotype in psoriatic patients was determined by flow cytometry analysis. We found the expansion of CD13^+^CD117^−^- and CD23^+^CD19^+^- (activated B) expressing cells in the patients with low A20 expression (Figure 1E,F), as compared to those with high A20 expression and healthy controls. Consistently, CD13 is well known to link to various inflammatory disorders [12] and an elevated number of CD23^+^ cells is found in atopic dermatitis and psoriasis [11]. In addition, the percentages of CD45^+^CD3^+^CD4^+^CD25^+^FoxP3^+^- (regulatory T), CD45^+^CD19^+^CD25^+^- (activated B), CD45^+^CD3^+^CD25^+^- (activated T) and CD13^+^CD117^+^-expressing cells were unaltered compared to healthy controls (data not shown). The results suggest that A20 expression in psoriatic patients was related to lipid metabolism, and the expansion of CD13^+^CD117^−^- and CD23^+^CD19^+^-expressing cells into circulation.

### 3.2. DNA Sequencing of JAK2 and DUB Genes in Psoriatic Patients

Firstly, sequencing of the JAK2 gene identified six nucleotide changes, including c.2109-142 G>A, rs974944169 T>A and rs994555780 T>C in intron 12, and rs4495487 T>C, rs372608048 C>G and rs10974947 G>A in intron 13 (Figure 2A, Table 2). Recently, polymorphisms in the JAK2 gene, including rs2274471 and rs7849191, have been detected in patients with psoriasis in the Korean population [21]. The genotype distribution of the six SNPs was in accordance with the Hardy–Weinberg equilibrium (HWE) (*p* > 0.05) (Table 3). Importantly, the TC genotype of the rs4495487 showed a higher frequency in the patient group (54.88%) compared to healthy individuals (40.8%, Table 2), while the GA genotype of the rs10974947 showed a lower frequency in the patient group (17.07%) compared to healthy individuals (36.05%, Table 2).

Next, the sequencing of the A20 gene identified two nucleotide changes, including rs745670694 G>A and rs200878487 C>G in exon 7 (Figure 2B, Table 2). The genotype distribution of the two SNPs was in accordance with the HWE (*p* > 0.05) (Table 3). Moreover, the CG genotype of the rs200878487 showed a higher frequency in the patient group (6.1%) compared to healthy individuals (0%; *p* = 0.029, Table 2). However, the effects of this SNP on protein function were not detected by Polyphen-2 (data not shown).

The sequencing of the CYLD gene identified that an SNP (c.2351-118) in intron 14 and two SNPs (c.2483+53 G>A and c.2483+188 G>A) in intron 15 were found (Figure 2C, Table 2). The genotype distribution of the three SNPs was in agreement with the HWE (*p* > 0.05) (Table 3).

Finally, the sequencing of the Cezanne gene identified eight nucleotide changes in intron 10; six out of the eight intronic SNPs (c.1584-437 T>A, c.1584-418 C>A, c.1584-375 A>C, c.1584-374 A>C, c.1584-278 G>A and c.1584-128 G>A) were unidentified SNPs and the two remaining intronic SNPs (rs587631702 A>T and rs1230581026 C>T) are reported in NCBI’s SNP database (Table 2, Figure 3A). Moreover, three exonic nucleotide changes, including rs1030371296 C>T, rs782178516 C>T and c.1642 T>A (*p*. W433R), in exon 11 were detected in this gene, in which the SNPs rs1030371296 C>T and c.1642 T>A (p.W433R) were non-synonymous SNPs (nsSNPs), causing changes in the amino acid residues, and a remaining SNP rs782178516 C>T was silent (Table 2, Figure 3A). The genotype distribution of the 11 SNPs in the Cezanne gene was in accordance with the HWE (*p* > 0.05, Table 3). Among them, the carrier frequencies of the four SNPs c.1584-375 A>C, c.1584-374 A>C, rs1230581026 C>T and p.W433R in psoriatic patients were significantly higher (14.63, 14.63, 21.95 and 6.1%, respectively) compared to healthy controls (0, 0, 0.68 and 0%, respectively, Table 2). In addition, the seven other SNPs in the Cezanne gene were not significantly associated with the psoriasis phenotype.

After the determination of susceptibility to psoriasis by evaluating the deleterious effects of the nsSNPs on the Cezanne gene, p.W433R was predicted to be probably damaging by Polyphen-2 with a score of 1 (score range: 0–1; sensitivity: 0; specificity: 1) (Figure 3B). Accordingly, p.W433R might be one of the most deleterious nsSNPs in the Cezanne gene. This reveals the association of p.W433R with susceptibility to the progression of psoriasis.

### 3.3. Haplotype and Linkage Disequilibrium Analysis of JAK2 and DUB Genes in Psoriatic Patients

Lastly, we tested the association of statistically inferred haplotypes with the risks of psoriasis. As shown in Figure 3C and Table 2, the SNPs in the Cezanne gene formed one haplotype block and contributed to two haplotypes in our study. This block was found to include the two SNPs c.1584-375c and c.1584-374. By using the common haplotype AA/AA as a reference, the haplotype AA/CC was associated with a high risk of psoriasis (14.63% for patients vs. 0% for controls, *p* < 0.001) (Table 2). The linkage disequilibrium (LD) analysis showed a tight linkage between the two SNPs c.1584-375c and c.1584-374 in the Cezanne gene. In addition, no haplotype blocks were found in JAK2, A20 or CYLD genes (data not shown).

## 4. Discussion

In this study, we observed for the first time that the expressions of A20, CYLD and Cezanne were inactivated in psoriatic patients. In agreement, A20 prevents inflammatory reactions in psoriatic arthritis-like disease [29], and the inactivation of A20 is related to psoriasis severity [18]. Unlike A20, the inactivation of CYLD is found in blood cancers, including leukemia [13], and a downregulated expression of Cezanne is reported in solid cancers, such as hepatocellular carcinoma [16].

To examine genetic alterations, we observed that the detected frequency of the CG genotype of rs200878487 in the A20 gene was significantly higher in the patient group compared to healthy individuals. Recently, two A20 variants, including rs582757 and rs6918329, have been identified as causal SNPs in psoriasis [30], and the c.1809delG SNP in exon 7 is linked to an early onset autoinflammatory syndrome [31]. A study on the Japanese population also indicated that several other SNPs in the A20 gene are associated with psoriasis susceptibility [17]. In mice, hypomorphic A20 expression makes one more susceptible to psoriasis-like skin inflammation [32]. Notably, in our study, rs200878487 in the A20 gene was not found in all the samples of the 70 patients with polycythaemia vera [33], or in the 155 patients with acute myeloid leukemia or the 110 patients with lymphocytic leukemia (unpublished data), suggesting that rs200878487 could be associated with a significant risk of psoriasis, but not blood cancers.

Next, CYLD is prominently expressed in the epidermis; however, CYLD^−/−^ mice do not exhibit skin abnormalities [34]. Recently, two SNPs (rs4785452 and rs12925755) in the CYLD gene have been reported to link to autoimmune diseases and psoriasis [20]. In this study, we did not find a contribution of the three SNPs in the CYLD gene to the risk of psoriasis.

Unlike A20 and CYLD, associations between polymorphisms in the Cezanne gene and autoimmune diseases are not well documented. Among the 11 SNPs identified in the Cezanne gene, we found that 4 SNPs, including the AC genotype of c.1584-375, AC genotype of c.1584-374, CT genotype of rs1230581026 and TA genotype of p.W433R, left one at increased risks of psoriasis. Differently, our recent study indicated that the c.1584-287 SNP in this gene is involved with the risk of polycythemia vera [33]. Moreover, p.W433R was found to be the most likely to exert deleterious effects, using the Polyphen-2 prediction tool. Interestingly, we demonstrated that the AA/CC haplotype in the Cezanne gene was detected at a higher frequency in patients with psoriasis compared to healthy controls (Table 2). The effect of this Cezanne haplotype in psoriasis susceptibility was reported for the first time here.

Functional studies of the A20 gene have demonstrated that A20 plays important roles in regulating pro-inflammatory gene expression through the activation of intracellular signaling pathways such as JAK2 [22]. Here, the frequency of the TC genotype of rs4495487 in the JAK2 gene was found to be significantly higher, while the GA genotype of rs10974947 in the JAK2 gene showed a lower frequency in the patient group compared to healthy individuals. Differently, rs4495487 SNP is associated with the JAK2-positive myeloproliferative neoplasm in the Japanese population [35], and the rs10974947 SNP in the JAK2 gene is involved only in polycythemia vera, and not other myeloproliferative disorders [36]. The rs7849191 SNP is strongly associated with the rs2274471 SNP in the JAK2 gene and a decreased risk of psoriasis; however, patients carrying the rs7849191 SNP tend to develop psoriasis at a late age in the Korean population [21].

For the determination of immunophenotypic features, we observed the expansion of CD13^+^CD117^−^- and CD23^+^CD19^+^-expressing cells in patients with a low A20 expression as compared to those with a high A20 expression and healthy controls. Consistently, CD13 is up-regulated on endothelial cells at sites of inflammation [37], and over-expressed by psoriatic fibroblasts [10]. The inhibition of CD13 results in impressive anti-inflammatory effects [12]. However, the associations between A20 expression levels and the accumulation of CD13^+^CD117^−^ and CD23^+^CD19^+^ cells in the circulation in patients with psoriasis were revealed for the first time. Recently, B cell-activation plays important roles in different pathological stages of psoriasis, and is correlated with disease severity [4,6], whereas the number of IL10-producing regulatory B cells is limited in patients with psoriasis [8]. We additionally observed that higher proportions of CD3^+^CD4^+^ and CD25^+^ T cells were found in four subjects, with an incidence of 4.88%, whereas the numbers of these cells in other patients were unchanged (data not shown). Previously, the activation of CD4 T cells in the peripheral blood of psoriatic patients has been determined to cause prolonged inflammation [7].

Similarly to a recent study [38], we observed that the levels of triglyceride and total cholesterol were significantly enhanced in psoriatic patients compared to healthy controls, and were even significantly higher in patients with low A20 expression levels compared to those with high A20 expression. In addition, the levels of CYLD and Cezanne did not affect lipid metabolism in these patients. A20 is considered to inhibit lipid accumulation and promote liver regeneration [39], and the overexpression of CYLD suppresses fatty acid synthesis in hepatocytes [40].

Lastly, the levels of CA125 were significantly increased in the patient group. Recently, CA125 has been found to be constitutively expressed in the skin of psoriatic patients [41], and its levels are elevated in patients with systemic lupus erythematosus and rheumatoid arthritis [27,28]. Interestingly, psoriatic patients with higher CA 125 levels than 35 U/mL showed increasing levels of IFN-γ, but not TNF-α. IFN-γ is secreted by activated T cells, and CA 125 is present on T cells [42], demonstrating that an enhanced expression of CA125 may be correlated with the secretion of IFN-γ and immune response in psoriasis patients.

## 5. Conclusions

A20 levels have been found to be associated with lipid metabolism, and the recruitment of CD13^+^ CD117^−^ and activated B cells into circulation, in psoriatic patients. Moreover, the deleterious effects of the p.W433R variant in the Cezanne gene may contribute to the risk of psoriasis, making it a good object of further study regarding its role in regulating the functional activation of immune cells in psoriatic patients.

## Figures and Tables

**Figure 1 medicina-59-01766-f001:**
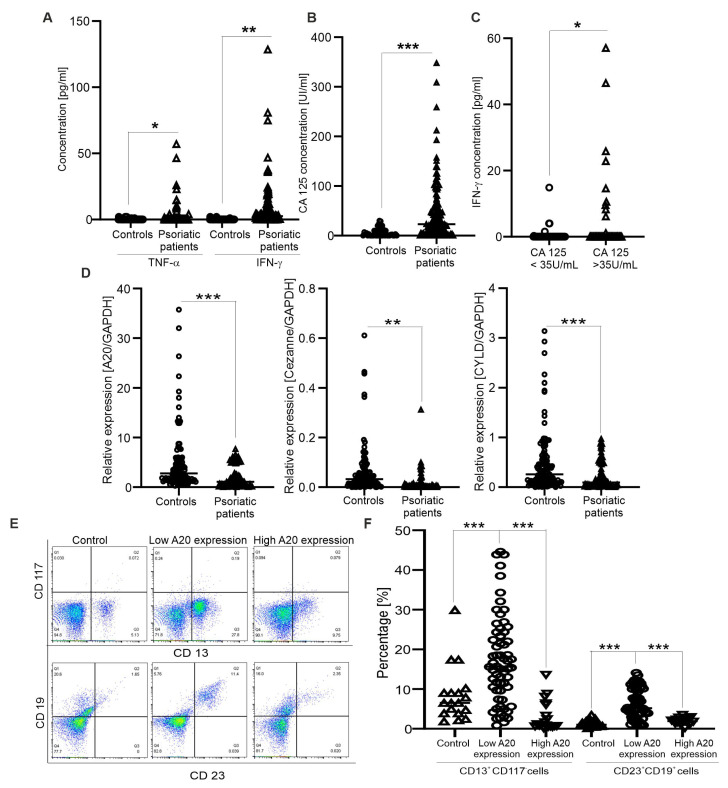
Correlation between expression of deubiquitinase genes and immunophenotypic features in psoriatic patients. (**A**,**B**) Graphs indicate TNF-α, INF-γ and CA 125 concentrations from sera of control individuals (*n* = 54) and psoriatic patients (*n* = 82); each dot represents a single sample. * (*p* < 0.05), ** (*p* < 0.01) and *** (*p* < 0.001) show significant differences from healthy individuals (Mann–Whitney U test). (**C**) Graph indicates INF-γ contents from sera of patients had higher/lower CA125 levels than 35 U/mL (*n* = 36–46). * (*p* < 0.05) shows significant difference from patients with normal CA125 levels (Mann–Whitney U test). (**D**) Graphs indicate the mRNA levels of A20, CYLD and Cezanne in psoriatic patients (*n* = 82) and control individuals (*n* = 88); GAPDH was used as a reference gene for relative quantification. ** (*p* < 0.01) and *** (*p* < 0.001) show significant differences from healthy individuals (Mann–Whitney U test). (**E**) Original dot plots of CD13^+^CD117^−^- and CD23^+^CD19^+^-expressing cells are shown for controls and psoriatic patients with low or high A20 expressions. (**F**) Graph indicates the percentages of CD13^+^CD117^−^- and CD23^+^CD19^+^-expressing cells (*n* = 20–62) in controls and psoriatic patients with low or high A20 expression. *** (*p* < 0.001) shows significant difference from patients with low A20 expression (Mann–Whitney U test).

**Figure 2 medicina-59-01766-f002:**
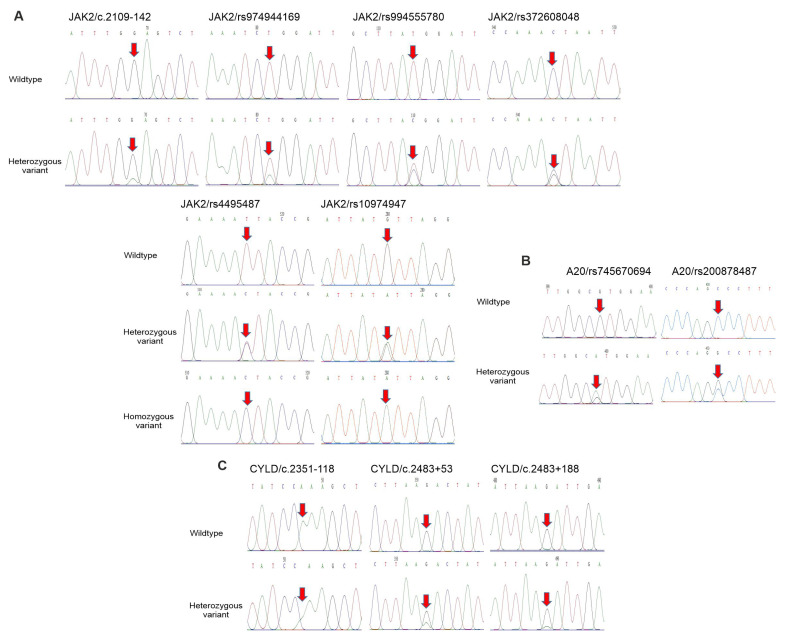
Polymorphisms of JAK2, A20 and CYLD genes in psoriatic patients and controls. (**A**–**C**) Partial sequence chromatograms of wildtype (1st panel) and variant (2nd and 3rd panels) genotypes of the SNPs c.2109-142, rs974944169, rs994555780, rs4495487, rs372608048 and rs10974947 of JAK2 gene (**A**), SNPs rs745670694 and rs200878487 of the A20 gene (**B**) and SNPs c.2351-118, c.2483+53 and c.2483+188 of the CYLD gene (**C**) are shown. Arrows indicate the locations of the base changes.

**Figure 3 medicina-59-01766-f003:**
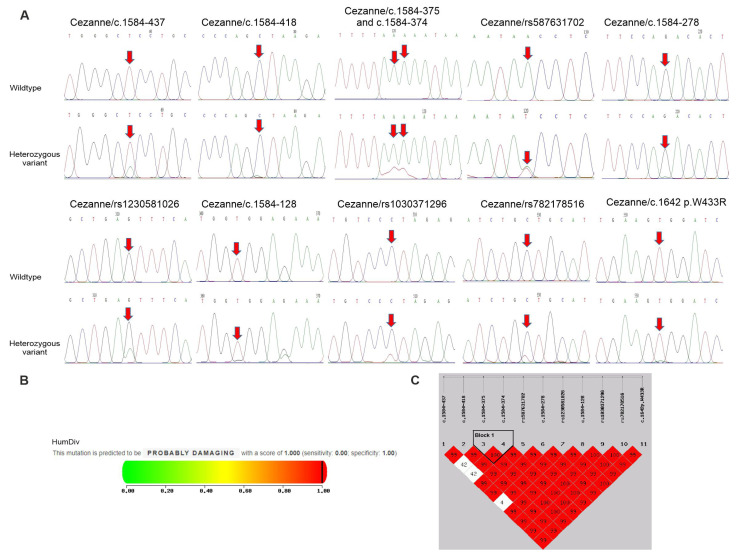
Polymorphisms of Cezanne gene in psoriatic patients and controls. (**A**) Partial sequence chromatograms of wildtype (1st panel) and variant (2nd panels) genotypes of the SNPs c.1584-437, c.1584-418, c.1584-375, c.1584-374, c.1584-278, c.1584-128, rs587631702, rs1230581026, rs1030371296, rs782178516 and p.W433R of the Cezanne gene are shown. Arrows indicate the locations of the base changes. (**B**) Functional prediction of the p. W433R variant by the Polyphen-2 program. (**C**) Linkage disequilibrium (LD) analysis shows the variants in the Cezanne gene and D’ values are shown in the LD blocks.

**Table 1 medicina-59-01766-t001:** Association between A20 expression and clinical parameters in psoriatic patients.

Characteristic	Normal Range	Total	A20		
Number of Patients		*n* = 82	Low (*n* = 62)	High (*n* = 20)	*p* Value
Age (years)		36.9 (22–70)	37.5 (24–70)	36.2 (22–65)	
Sex, male (n, %)		56 (68%)	45 (73%)	11 (55%)	
Urea (mmol/L)	3.3–6.6	4.76 ± 1.05	4.75 ± 0.99	4.8 ± 1.26	0.85
Glucose (mmol/L)	3.9–5.6	5.48 ± 0.65	5.62 ± 0.57	5.31 ± 0.62	0.808
Creatinine (µmol/L)	50–110	83.38 ± 12.32	83.55 ± 17.4	82.7 ± 22.04	0.854
Triglyceride (mmol/L)	<1.7	**2.54 ± 0.81**	2.65 ± 0.85	2.13 ± 0.47	**0.01 ****
Total cholesterol (mmol/L)	<5.2	**5.53 ± 1.18**	5.7 ± 1.22	4.84 ± 0.63	**0.0029 *****
LDL-Cholesterol (mmol/L)	<3.4	2.95 ± 0.80	2.98 ± 0.85	2.91 ± 0.74	0.746
HDL-Cholesterol (mmol/L)	>0.9	1.11 ± 0.32	1.115 ± 0.286	1.1 ± 0.33	0.603
AST (U/L)	5–40	24.92 ± 4.63	25.12 ± 5.14	24. 09± 5.03	0.424
ALT (U/L)	7–55	25.23 ± 5.29	27.12 ± 11.54	25.66 ± 5.98	0.586
GGT (UI/L)	<66	59.12 ± 8.19	60.03 ± 8.56	58.81 ± 9.11	0.224

*n* = number of subjects, low density lipoprotein: LDL, high density lipoprotein: HDL, aspartate aminotransferase: AST, alanine Aminotransferase: ALT, gamma glutamyl transferase: GGT, ** (*p* < 0.01) and *** (*p* < 0.001) show significant differences between the low and high A20 expression groups (Mann–Whitney U test), bold is used to distinguish the higher clinical parameters than normal values.

**Table 2 medicina-59-01766-t002:** Comparison of genotype and haplotype frequencies of JAK2 and deubiquitinase genes between psoriatic patients and controls.

SNP	Gene	Test Model	Controls (*n* = 147)	Psoriatic Patients (*n* = 82)	OR	95% CI	*p*-Value
Genotype							
c.2109-142	JAK2	GG	147 (100%)	81 (98.78%)	1		
		GA	0 (0%)	1 (1.22%)	5.4294	0.2187–134.8167	1 ^(1)^
rs974944169	JAK2	TT	147 (100%)	81 (98.78%)	1		
		TA	0 (0%)	1 (1.22%)	5.4294	0.2187–134.8167	1 ^(1)^
rs994555780	JAK2	TT	145 (98.64%)	80 (97.56%)	1		
		TC	2 (1.36%)	2 (2.44%)	1.7901	0.2475–12.9494	1 ^(1)^
rs4495487	JAK2	TT	66 (44.9%)	25 (30.5%)	1		
		TC	60 (40.8%)	45 (54.88%)	1.98	1.0855–3.6116	**0.046 ^(2)^**
		CC	21 (14.3%)	12 (14.62%)	1.5086	0.6477–3.5138	0.381 ^(1)^
rs372608048	JAK2	CC	147 (100%)	80 (97.56%)	1		
		CG	0 (0%)	2 (2.44%)	9.1615	0.4345–193.1661	0.497 ^(1)^
rs10974947	JAK2	GG	90 (61.22%)	65 (79.27%)	1		
		GA	53 (36.05%)	14 (17.07%)	0.3657	0.1872–0.7146	**0.004 ^(2)^**
		AA	4 (2.73%)	3 (3.66%)	1.0385	0.2247–4.7987	1 ^(1)^
rs745670694	A20	GG	142 (96.7%)	82 (100%)	1		
		AG	5 (3.4%)	0 (0%)	0.157	0.0086–2.8761	0.246 ^(1)^
rs200878487	A20	CC	147 (100%)	77 (93.9%)	1		
		CG	0 (0%)	5 (6.1%)	20.9355	1.1426–383.5949	**0.029 ^(1)^**
c.2351-118	CYLD	AA	139 (94.56%)	80 (97.56%)	1		
		DelA	8 (5.44%)	2 (2.44%)	0.4344	0.0900–2.0957	0.445 ^(1)^
c.2483+53	CYLD	GG	110 (74.83%)	59 (71.95%)	1		
		GA	37 (25.17%)	23 (28.05%)	1.159	0.6303–2.1309	0.629 ^(2)^
c.2483+188	CYLD	GG	143 (97.28%)	78 (95.12%)	1		
		GA	4 (2.72%)	4 (4.88%)	1.8333	0.4462–7.5327	0.721 ^(1)^
c.1584-437	Cezanne	TT	137 (91.2%)	78 (95.12%)	1		
		TA	10 (8.8%)	4 (4.88%)	0.7026	0.2132–2.3150	0.407 ^(1)^
c.1584-418	Cezanne	CC	147 (100%)	80 (97.56%)	1		
		CA	0	2 (2.44%)	9.1615	0.4345–193.1661	0.497 ^(1)^
c.1584-375	Cezanne	AA	147 (100%)	70 (85.37%)	1		
		AC	0	12 (14.63%)	52.305	3.0530–896.1009	**<0.001 ^(1)^**
c.1584-374	Cezanne	AA	147 (100%)	70 (85.37%)	1		
		AC	0	12 (14.63%)	52.305	3.0530–896.1009	**<0.001 ^(1)^**
rs587631702	Cezanne	AA	147 (100%)	81 (98.78%)	1		
		AT	0	1 (1.22%)	5.4294	0.2187–134.8167	1^(1)^
c.1584-278	Cezanne	GG	145 (98.64%)	79 (96.34%)	1		
		GA	2 (1.36%)	3 (3.66%)	2.7532	0.4505–16.8247	0.369 ^(1)^
rs1230581026	Cezanne	CC	146 (99.32%)	64 (78.05%)	1		
		CT	1 (0.68%)	18 (21.95%)	41.0625	5.3659–314.2310	**<0.001 ^(1)^**
c.1584-128	Cezanne	GG	147 (100%)	80 (97.56%)	1		
		GA	0	2 (2.44%)	9.1615	0.4345–193.1661	0.497 ^(1)^
rs1030371296	Cezanne	CC	147 (100%)	81 (98.78%)	1		
		CT	0	1 (1.22%)	5.4294	0.2187–134.8167	1^(1)^
rs782178516	Cezanne	CC	147 (100%)	81 (98.78%)	1		
		CT	0	1 (1.22%)	5.4294	0.2187–134.8167	1^(1)^
c.1642 p. W433R	Cezanne	TT	147 (100%)	78 (93.9%)	1		
		TA	0	5 (6.1%)	20.6688	1.1282–378.6672	**0.029 ^(1)^**
**Haplotype block**							
c.1584-375c andc.1584-374	Cezanne	AA/AA	147 (100%)	70 (85.37%)	1		
		AA/CC	0	12 (14.63%)	52.305	3.0530–896.1009	**<0.001 ^(1)^**

*p*-values were calculated by either Fisher’s exact test ^(1)^ or Chi-squared test ^(2)^; *p* < 0.05 (in bold) indicates statistical significance from healthy donors; OR: odds ratio; 95% CI: 95% confidence interval of odds ratio.

**Table 3 medicina-59-01766-t003:** General information on SNPs of JAK2 and deubiquitinase genes in psoriatic patients and controls.

Gene/SNP	Type of Variant	Allele	MAF		HWE (*p*-Value)		
			Controls	Patients	Controls	Patients	All Population
JAK2/c.2109-142	Intron	G>A	0.0000	0.0061	N/A	0.9985	0.9995
JAK2/rs974944169	Intron	T>A	0.0000	0.0061	N/A	0.9985	0.9995
JAK2/rs994555780	Intron	T>C	0.0068	0.0122	0.9966	0.9938	0.9911
JAK2/rs4495487	Intron	T>C	0.3469	0.4207	0.4847	0.5223	0.9548
JAK2/rs372608048	Intron	C>G	0.0000	0.0122	N/A	0.9938	0.9978
JAK2/rs10974947	Intron	G>A	0.2075	0.1220	0.5057	0.1853	0.9973
A20/ rs745670694	Synonymous	G>A	0.0170	0.0000	0.9782	N/A	0.9861
A20/ rs200878487	Missense	C>G	0.0000	0.0305	N/A	0.9603	0.9861
CYLD/ c.2351-118	Intron	Del A	0.0272	0.0122	0.9441	0.9938	0.1141
CYLD/c.2483+53	Intron	G>A	0.1259	0.1402	0.218	0.3359	0.07411
CYLD/c.2483+188	Intron	G>A	0.0136	0.0244	0.9861	0.9747	0.9646
Cezanne/c.1584-437	Intron	T>A	0.0340	0.0244	0.9129	0.9747	0.8924
Cezanne/c.1584-418	Intron	C>A	0.0000	0.0122	N/A	0.9938	0.9978
Cezanne/c.1584-375	Intron	A>C	0.0000	0.0732	N/A	0.7745	0.9204
Cezanne/c.1584-374	Intron	A>C	0.0000	0.0732	N/A	0.7745	0.9204
Cezanne/rs587631702	Intron	A>T	0.0000	0.0061	N/A	0.9985	0.9995
Cezanne/c.1584-278	Intron	G>A	0.0068	0.0183	0.9966	0.9859	0.9861
Cezanne/ rs1230581026	Intron	C>T	0.0034	0.1098	0.9992	0.5362	0.8069
Cezanne/c.1584-128	Intron	G>A	0.0000	0.0122	N/A	0.9938	0.9978
Cezanne/rs1030371296	Missense	C>T	0.0000	0.0061	N/A	0.9985	0.9995
Cezanne/rs782178516	Synonymous	C>T	0.0000	0.0061	N/A	0.9985	0.9995
Cezanne/c.1642 p.W433R	Missense	T>A	0.0000	0.0305	N/A	0.9607	0.9862

Position refers to the GRCh38.p10 assembly; MAF: minor allele frequency; HWE: Hardy–Weinberg equilibrium was checked by Chi-squared test; N/A: not available.

## Data Availability

The data that support the findings of this study are available from the corresponding author upon reasonable request.

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
