# Peer review of "Associations of A20, CYLD, Cezanne and JAK2 Genes and Immunophenotype with Psoriasis Susceptibility"

_medicina, 2023, doi:10.3390/medicina59101766_

Round 1

Reviewer 1 Report

The article is well written, the information makes an important contribution to the study of psoriasis. There are some disagreements in the text that need to be corrected.  

Author Response

The article is well written, the information makes an important contribution to the study of psoriasis. There are some disagreements in the text that need to be corrected. 

 Response 1: Thank you for the comments. We improved English grammar in the paper by a native speaker and corrected several mistakes as shown in highlighted texts.

Reviewer 2 Report

The article presents very interesting results, which, if confirmed by further research, could contribute to further knowledge of the immunological/biological nature of psoriasis.

However, despite the interesting results, a number of things need to be revised:

• A number of theoretical assumptions are presented as clear established facts, although many are still hypothetical, based on only a very limited number of studies, in some cases even just one (e.g. CA125 expression in psoriatic skin...role of CA125 in autoimmune diseases....involvement of B-lymphocytes in psoriasis, etc.). In all such cases, this needs to be reflected and better presented/explained, including how much data is currently available on such assumptions.

• Contrary to the previous, the statement that psoriasis „may be“ associated with abnormalities/disorders in lipid metabolism is not accurate because dyslipidemia is an established comorbidity of psoriasis.

• For CD23, do not refer to allergic rhinitis (reference 12) because it is a TH2 type of inflammation and it has nothing to do with psoiasis.

• It is definitely not true that tofacitinib and ruxolitinib are used in the treatment of psoriasis (reference 22), both drugs have eventually failed in the clinical trials!

• All figures must be larger/more virble

• Widen Table 1 so that all numbers are on just one line and add a legend/explanation for what ** and *** mean.

• Unfortunately, the level of English is very low, both in terms of stylistics and a number of basic grammatical errors. Please hire a professional English editor to correct the article for you.

See above

Author Response

The article presents very interesting results, which, if confirmed by further research, could contribute to further knowledge of the immunological/biological nature of psoriasis.

However, despite the interesting results, a number of things need to be revised:

  • A number of theoretical assumptions are presented as clear established facts, although many are still hypothetical, based on only a very limited number of studies, in some cases even just one (e.g. CA125 expression in psoriatic skin...role of CA125 in autoimmune diseases....involvement of B-lymphocytes in psoriasis, etc.). In all such cases, this needs to be reflected and better presented/explained, including how much data is currently available on such assumptions.

Response 1: Yes, we added more references and made several sentence corrections in the dicussion section as shown belows:

- “CA125 levels are elevated in patients with systemic lupus erythematosus and rheumatoid arthritis [27,28]”

- “B cells activation play important roles in different pathological stages of psoriasis and correlate with disease severity [4,6], whereas the number of IL10-producing regulatory B cells are impaired in patients with psoriasis [8]”

 Contrary to the previous, the statement that psoriasis „may be“ associated with abnormalities/disorders in lipid metabolism is not accurate because dyslipidemia is an established comorbidity of psoriasis.

Response 2: Yes, we deleted the sentence “psoriasis may be associated with lipid metabolism disorders” as suggested

  • For CD23, do not refer to allergic rhinitis (reference 12) because it is a TH2 type of inflammation and it has nothing to do with psoiasis.

Response 3: Yes, we deleted the reference 12 as suggested

  • It is definitely not true that tofacitinib and ruxolitinib are used in the treatment of psoriasis (reference 22), both drugs have eventually failed in the clinical trials!

Response 4: Yes, we deleted the reference 22 as suggested

  • All figures must be larger/more visible

Response 5: Yes, we made all figures larger/more visible as required

  • Widen Table 1 so that all numbers are on just one line and add a legend/explanation for what ** and *** mean.

Response 6: Yes, Table 1 is now widen and we added a explanation for what the ** and *** mean.

  • Unfortunately, the level of English is very low, both in terms of stylistics and a number of basic grammatical errors. Please hire a professional English editor to correct the article for you.

Response 7: Yes, we improved English grammar in the paper by a native speaker as required

Reviewer 3 Report

Deer Authors,

The manuscript presented are a good attempt to uncover the new aspects of psoriasis pathogenesis related deubiquitinase genes. Despite good scientific merit of the manuscript, there are some issues that have to be addressed befor the considering for publication.

First, Change the Title of the manuscript. I wonder if the miscorrect reflection of the manuscript content is arose from not so good English? Anyway, You have to clearly reflect the aim of the research in the Title.

Second, there are association found between eg., the frequency of a SNP polymorphism and psoriatic patients cohort. Did you made an association analysis or just calculate statistic probability of difference between healthy and patients? Please, indicate the software for association analysis used, in Materials and methods. Besides, if you analysis several SNP in a gene, why are you not calculate linkage disequillirium value. You are free to use 'hapmap' for the best presentation of results.

The quality of English are not so good and the fact often leads to sense misunderstanding.

Author Response

The manuscript presented are a good attempt to uncover the new aspects of psoriasis pathogenesis related deubiquitinase genes. Despite good scientific merit of the manuscript, there are some issues that have to be addressed before the considering for publication.

First, Change the Title of the manuscript. I wonder if the miscorrect reflection of the manuscript content is arose from not so good English? Anyway, You have to clearly reflect the aim of the research in the Title.

Response 1: Yes, the title of the manuscript is changed as suggested

Second, there are association found between eg., the frequency of a SNP polymorphism and psoriatic patient cohort. Did you made an association analysis or just calculate statistical probability of difference between healthy and patients? Please, indicate the software for association analysis used, in Materials and methods. Besides, if you analysis several SNP in a gene, why are you not calculate linkage disequillirium value. You are free to use 'hapmap' for the best presentation of results.

Response 2:

- The frequency of SNP polymorphisms and psoriatic patients is calculated by statistical probability of difference between healthy and patients.

- We added texts in the Method section as follows: “To analyze the data chi-square (χ2) tests, crosstabs and odds ratios were used for dichotomous data and the Mann-Whitney U test for nominal data”.

- The linkage disequillirium analysis is indicated in Figure 3C.
